# Dual-channel Dynamic Graph Neural Networks with Adaptive Adjacency Learning and Multi-scale Representation Fusion

**Youqing Wang** [1]  **Jiahao Long** [1]  **Tianxiang Zhao** [1]  **Man Cao** [1]  **Mengyuan Xin** [1]  **Jiapu Wang** [2]  **Junbin Gao** [3]
**Jipeng Guo** [1]

## Abstract

Graph neural networks (GNNs) have been demonstrated to be powerful tools for analyzing structural graph data. However, most existing methods usually rely on fixed adjacency structures for information propagation, lacking strong adaptability to the latent semantic relationships that exist but are not explicitly connected in graph, especially in complementary high-pass and low-pass filtering views. To this end, this paper proposes a novel Dual-channel Dynamic Graph Neural Network (DCD-GNN), mainly consisting of parallel representation learning channels: a static structure-preserving channel and a dynamic adjacency-enhancing channel. The dynamic channel exploits both low-pass structural filtering and high-pass personalized detail via the self-attention adjacency learning and then integrates them for comprehensive semantic modeling, while the static channel maintains structural stability. Both channels employ a multi-scale representation fusion mechanism and a unified and discriminative node embedding representation is obtained by integrating them. Extensive experiments on various benchmark datasets verify the superiority of DCD-GNN in discriminative graph representation learning.

## 1. Introduction

Graph data are widely present in real-world scenarios, such as social networks (Li et al., 2023b), citation networks (Cummings & Nassar, 2020), chemical molecular structures (Wu et al., 2023), and financial transaction networks (Cui et al., 2021). The complex relational patterns inherent in such data present both significant opportunities and challenges for data mining and analysis. As a universal data structure, graphs effectively capture intricate relationships among entities across various domains. In recent years, Graph Neural Networks (GNNs) have attracted widespread attention due to their discriminative representation learning ability for graph-structured data, achieving notable success in various graph tasks including node classification (Wu et al., 2021; Wang et al., 2019; Jiang et al., 2026), link prediction (Xie et al., 2022), traffic forecasting (Guo et al., 2020), clustering (Guan et al., 2025; Yang et al., 2026a), and recommendation systems (Li et al., 2023a).

In general, most existing GNNs are primarily built upon the message passing framework, whose core idea is to learn node embedding by iteratively aggregating neighboring information (Wang et al., 2026; Yang et al., 2024). For example, ChebNet (He et al., 2022) employs Chebyshev polynomials as simplified convolutional filters; GCN (Kipf & Welling, 2017) further simplifies ChebNet to a first-order approximation and has become a representative method due to its powerful ability to propagate and aggregate features over graphs. PPNP (Klicpera et al., 2019) and its efficient approximation version APPNP (Klicpera et al., 2019) introduce a propagation mechanism based on personalized PageRank, which decouples feature learning from graph propagation and establishes connections between input and output to enhance node importance. Methods such as GCNII (Chen et al., 2020) and DAGNN (Liu et al., 2020) alleviate over-smoothing and gradient degradation issues in deep GNNs by incorporating techniques like initial residual connections and identity mapping. GNN-BC (Yang et al., 2022) adopts dual independent representation learning for attributes and topology to mitigate mutual interference and redundancy between them. NoSAF and NoSAF-D (Wang et al., 2024) further propose layer-aggregation and filtering architectures with node-specific layers, handling heterogeneous graph data effectively and enhancing network depth and performance through per-node information filtering and dynamic codebook integration. TFE-GNN (Duan et al., 2024) draws inspiration from ensemble Learning theory, extracting ro-

---

[1]College of Information Science and Technology, Beijing University of Chemical Technology [2]Nanjing University of Science and Technology [3]The University of Sydney. Correspondence to: **Jipeng Guo** and **Tianxiang Zhao** <guojipeng@buct.edu.cn, zhaotianxiang@buct.edu.cn>.

*Proceedings of the $43^{rd}$ International Conference on Machine Learning*, Seoul, South Korea. PMLR 306, 2026. Copyright 2026 by the author(s).

bust classifiers from base classifiers via a triple-filter mechanism that unifies homophilic and heterophilic patterns in spectral graph neural networks. EvenNet (Lei et al., 2022) discards odd-hop neighbors and improves model robustness via even-polynomial graph filters.

Despite the significant advances, most existing methods excessively rely on the original static graph structure. Whether based on convolutional or message-passing paradigms, most existing approaches restrict information propagation strictly to the predefined topology, failing to adequately explore implicit semantic and structural relationships in the graph. Traditional GNNs typically assume that the input adjacency matrix is static and perfectly accurate, yet real-world graphs often suffer from incompleteness, noise, and even dynamic evolution. For instance, in social networks, implicit interest associations between users often go beyond explicit follow relationships; in citation networks, cross-domain thematic connections may not be fully captured by existing citation edges. If the expressive power of GNNs is confined only to explicit connections, many implicit semantic associations that are crucial for downstream tasks could be ignored. Moreover, while some methods such as LHS (Qiu et al., 2024) attempt to enhance GNNs by automatically learning potential homophilic structures on heterophilic graphs, they still lack a systematic integration of full-frequency information, making it difficult to comprehensively capture the complex latent semantics and structural relationships.

To comprehensively and effectively explore latent semantic and structural information, this study proposes a novel framework termed the Dual-Channel Dynamic Graph Neural Network (DCD-GNN). Its core idea is to learn dynamic adaptability and structural stability separately through a dual-channel parallel architecture, enabling optimal fusion of multi-level and multi-scale structural information. Specifically, the dynamic channel consists of two complementary high-frequency and low-frequency sub-channels and utilizes the self-attention mechanism to explore multi-level semantic correlations among all nodes. Then, the low-pass structural filtering and high-pass details are integrated by adaptive fusion to fully capture latent semantic and structural patterns across the entire frequency spectrum. The static channel, on the other hand, preserves structural stability, extracting explicit topological information while preventing excessive dynamic adjustments that may cause structural distortion. Furthermore, DCD-GNN adopts a multi-scale representation fusion mechanism to fully account for the contribution of embeddings at different scales, which enables the model to adaptively determine information reliance based on specific graph structures and tasks. The main contributions of this paper are summarized as follows:

- The DCD-GNN explores both dynamic and static semantic patterns in a parallel way and integrates multi-scale representations in each channel, resulting in comprehensive multi-level and multi-scale representations.

- The dynamic channel captures implicit semantic correlations by self-attention learning in both high-pass and low-pass filtering, endowing the proposed DCD-GNN with strong representation learning flexibility.

- Extensive experiments conducted on real-world benchmark datasets verify that DCD-GNN outperforms state-of-the-art methods in classification tasks.

## 2. Preliminaries

Given a graph $\mathcal{G} = (\mathbf{X}, \mathcal{V}, \mathcal{E})$, $\mathcal{V} = \{v_1, \cdots, v_N\}$ is node set with $N$ sample nodes, $\mathbf{X} \in \mathbb{R}^{N \times d}$ is attribute feature matrix of nodes, $\mathcal{E}$ is the connection edge set. Its topological structure can be represented by the adjacency matrix $\mathbf{A} = [a_{ij}] \in \{0,1\}^{N \times N}$, where $a_{ij} = 1$ if there is an edge $e_{ij} = (v_i, v_j) \in \mathcal{E}$ between node $v_i$ and node $v_j$, and $a_{ij} = 0$ otherwise. The GNNs aim at learning discriminative embedding representation by simultaneously utilizing attribute feature and graph structure. The graph convolutional operator is widely used to obtain graph embedding representation by the following explicit way

$$\mathbf{H}^{(l+1)} = \sigma\left(\widetilde{\mathbf{D}}^{-\frac{1}{2}} \widetilde{\mathbf{A}} \widetilde{\mathbf{D}}^{-\frac{1}{2}} \mathbf{H}^{(l)} \mathbf{W}^{(l)}\right) \tag{1}$$

where $\widetilde{\mathbf{A}} = \mathbf{A} + \mathbf{I}$ is the revised adjacency matrix of graph $\mathcal{G}$ with self-loops, $\mathbf{I}$ is an identity matrix, $\widetilde{\mathbf{D}}$ is a diagonal degree matrix with diagonal element $\widetilde{\mathbf{D}}_{ii} = \sum_j \widetilde{\mathbf{A}}_{ij}$. $\mathbf{H}^{(l+1)}$ and $\mathbf{H}^{(l)}$ are embedding representations of $(l+1)$-th and $l$-th layers, $\mathbf{W}^{(l)}$ is the trainable parameter matrix, and $\sigma(\cdot)$ is the activation function. Specifically, $\mathbf{H}^{(0)} = \mathbf{X}$ is defined.

GNNs with explicit graph convolution operators have been widely adopted in various tasks and have achieved satisfactory performance. Their essence lies in simultaneously leveraging the original topological structure and attribute features to learn embedding representations. However, this reliance on explicit structures constitutes a major bottleneck: model performance is highly constrained by the completeness and accuracy of the explicit graph topology. When the graph contains noisy connections, missing edges, or structural bias, the neighboring aggregation based on such structures inevitably introduces errors, resulting in distorted node representations and inevitably weakening the model's generalization ability in the complex real-world scenarios. Consequently, effectively modeling and integrating the latent semantic and relational information that is not explicitly expressed in the graph becomes crucial for enhancing both the robustness and the expressive power of GNNs (Yang et al., 2026b; Wang et al., 2025).

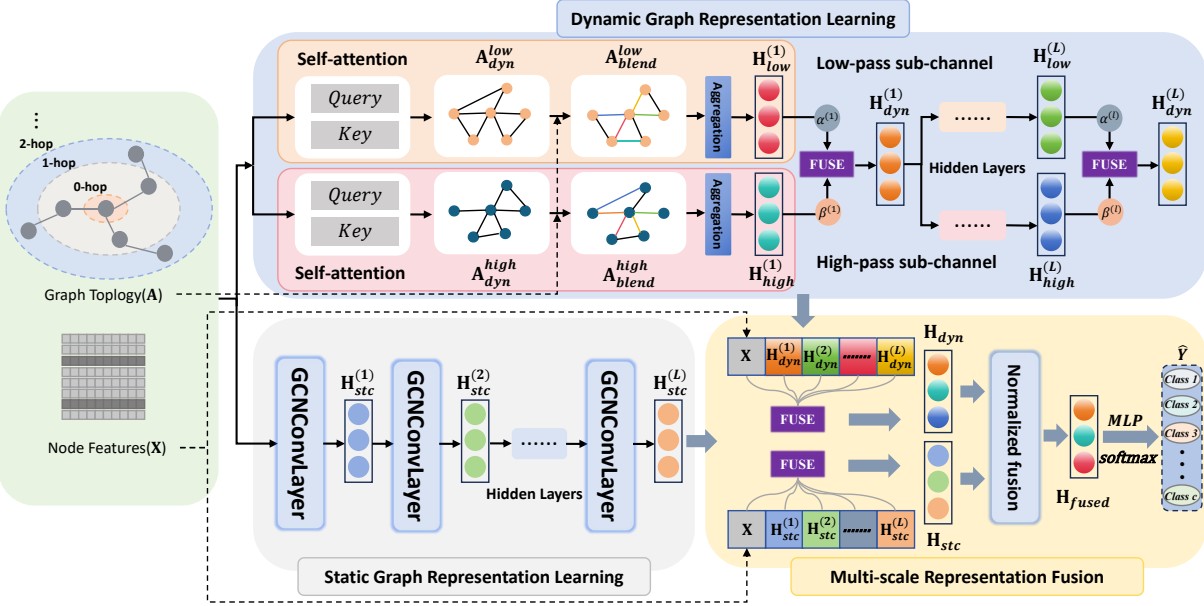

*Figure 1.* An overview of the proposed DCD-GNN model, which mainly consists of dynamic graph representation learning (DGRL), static graph representation learning (SGRL), and multi-scale representation fusion (MsRF) modules. The DGRL explores dynamic latent semantic connections from both low-pass and high-pass filtering. The SGRL module learns the stable embedding representation on original topological graph to provide representation foundation. And, the MsRF module integrates multi-level and multi-scale representations for comprehensive and discriminative embedding.

## 3. Methodology

Here, the details of the proposed DCD-GNN method are elaborated, where the architecture is sketched in Figure 1.

### 3.1. Overview of the Proposed DCD-GNN

To effectively improve the potential semantic learning capability of GNNs, this study proposes the DCD-GNN, which mainly comprises three key modules: 1) The **dynamic graph representation learning** (DGRL) module employs a self-attention learning mechanism to infer implicit inter-node relationships from node embeddings. It constructs a hybrid low-pass filtering adjacency matrix that emphasizes local smoothness and a hybrid high-pass filtering adjacency matrix that emphasizes local variability, thereby modeling homophily aggregation and heterophily relationships, respectively. 2) The **static graph representation learning** (SGRL) module aims to stably capture the inherent and explicit structural information in the original comprise graph, maintaining the representation learning stability. 3) The **multi-scale representation fusion** (MsRF) module adaptively fuses learned embedding representations from different propagation layers, enabling the DCD-GNN to effectively capture multi-scale semantic information. Ultimately, the representations from the dynamic and static branches are unified for downstream classification tasks.

### 3.2. Dynamic Graph Representation Learning

Real-world graphs often contain noisy connections or lack latent semantic associations. Static adjacency graph struggles to accurately reflect authentic node relationships, and the fixed topological structures fail to adapt to the continuously evolving high-order semantic dependencies across different network layers. The DGRL module, capable of capturing rich latent semantic and structural information, effectively mitigates noise interference and infers potential semantic relationships by adaptive semantic adjacency learning from both low-pass and high-pass filtering. Thus, it could better support multi-scale and multi-granularity graph representation learning demands.

**Adaptive Self-Attention Adjacency Learning** It adaptively reconstructs semantic adjacency relationships for each node to capture latent associations and reduce noise interference in the original graph. Due to the static and predefined adjacency matrix, information can only be aggregated from a fixed neighborhood, preventing access to global information from non-adjacent nodes. However, as real-world graphs often exhibit complex latent semantic and structural relationships that cannot be fully captured by a static adjacency matrix, many potential connections between nodes remain unexpressed despite the absence of

direct links. For example, in the ACM computing classification, where "Natural Language Processing" serves as the category of "Machine Translation" without an explicit connection, their initial embeddings reside far apart in the low-dimensional representation space. Drawing inspiration from the self-attention mechanism in Transformer (Vaswani et al., 2017), a self-attention adjacency learning is designed to compute attention correlation scores for each node pair, enabling end-to-end learning of a complementary adjacency structure derived dynamically from node embedding representation. Specifically, given the node representations at $l$-th layer, denoted as $\mathbf{H}^{(l)} \in \mathbb{R}^{N \times d}$, it is projected into query and key representations via linear transformations:

$$\mathbf{Q}^{(l)} = \mathbf{H}^{(l)}\mathbf{W}_{query}, \quad \mathbf{K}^{(l)} = \mathbf{H}^{(l)}\mathbf{W}_{key} \qquad (2)$$

where $\mathbf{W}_{query} \in \mathbb{R}^{d \times d}$ and $\mathbf{W}_{key} \in \mathbb{R}^{d \times d}$ are trainable transformation matrices. Subsequently, the fully connected relational weight matrix is computed via query-key dot-product similarity and normalized by the softmax function, obtaining the adaptive semantic adjacency matrix as follows:

$$\mathbf{A}_{dyn}^{(l)} = \mathrm{softmax}\left(\frac{\mathbf{Q}^{(l)}(\mathbf{K}^{(l)})^\top}{\sqrt{d}}\right) \qquad (3)$$

Here, dividing by $\sqrt{d}$ is applied for gradient stabilization. Each element $\mathbf{A}_{dyn}^{(l)}(i, j)$ can be interpreted as the attention correlation coefficient of node $v_i$ toward node $v_j$, encoding an implicit semantic relationship based on the current node embedding representations. Notably, $\mathbf{A}_{dyn}^{(l)}(i, j)$ is generally not equal to $\mathbf{A}_{dyn}^{(l)}(j, i)$, a pattern commonly observed in real-world graphs. For instance, the attention a student pays to a teacher is typically much higher than that paid by the teacher to the student. The constructed adaptive semantic adjacency $\mathbf{A}_{dyn}^{(l)}$ thus models unique semantic dependencies for the information flow of each node. To leverage the learned dynamic structure without excessively deviating from the basic topology of the original graph, a convex combination between the original adjacency matrix and the reconstructed adjacency matrix is performed to form comprehensive adjacency relationships, thereby balancing structural priors with dynamic adaptability:

$$\mathbf{A}_{blend}^{(l)} = \lambda\mathbf{A} + (1 - \lambda)\mathbf{A}_{dyn}^{(l)} \qquad (4)$$

where $\mathbf{A}$ represents the predefined static original adjacency matrix, $\lambda$ is a specific scalar parameter that controls the relative importance between the original structure and the dynamically reconstructed structure during information propagation, and $\mathbf{A}_{blend}^{(l)}$ denotes the final comprehensive adjacency matrix beyond the original physical connections.

**Graph Representation Learning with Adaptive Adjacency on Both Low-pass and High-pass Filtering.** In graph signals, low-pass filtering emphasizes smoothness between connected nodes, which may lead to over-smoothing of graph representation learning. To capture more comprehensive graph signal characteristics, both low-pass and high-pass views for graph filtering are explicitly constructed and fused. The low-pass filtering operator is defined as $\mathcal{A}^{low} = \widetilde{\mathbf{D}}^{-\frac{1}{2}}\widetilde{\mathbf{A}}\widetilde{\mathbf{D}}^{-\frac{1}{2}}$, where $\widetilde{\mathbf{A}}$ is any adjacency matrix that includes self-connections, $\widetilde{\mathbf{D}}$ is the degree diagonal matrix of $\widetilde{\mathbf{A}}$. For the high-pass filtering, a simple form is utilized, i.e., $\mathcal{A}^{high} = \mathbf{I} - \mathcal{A}^{low}$, which reflects node-specific information that differs from its neighbors. Specifically, for the low-pass and high-pass sub-channels in the dynamic channel, the information propagation rules at the $l$-th layer are respectively defined as:

$$\mathbf{H}_{low}^{(l)} = \mathrm{ReLU}\left(\mathcal{A}_{blend}^{low,(l)}\mathbf{H}_{dyn}^{(l)}\boldsymbol{\Theta}_{low}^{(l)}\right) \qquad (5)$$

$$\mathbf{H}_{high}^{(l)} = \mathrm{ReLU}\left(\mathcal{A}_{blend}^{high,(l)}\mathbf{H}_{dyn}^{(l)}\boldsymbol{\Theta}_{high}^{(l)}\right) \qquad (6)$$

where $\mathcal{A}_{blend}^{low,(l)}$ and $\mathcal{A}_{blend}^{high,(l)}$ represent the low-pass filtering and high-pass filtering operators with dynamic adjacency $\mathbf{A}_{blend}^{(l)}$ on the $l$-th layer, respectively. And, $\boldsymbol{\Theta}_{low}^{(l)}$ and $\boldsymbol{\Theta}_{high}^{(l)}$ are trainable weight matrices used for feature transformation. At each layer, the representations from the low-pass and high-pass views are integrated to form the output of the dynamic channel for that layer:

$$\mathbf{H}_{dyn}^{(l+1)} = \alpha^{(l)}\mathbf{H}_{low}^{(l)} + \beta^{(l)}\mathbf{H}_{high}^{(l)} \qquad (7)$$

where $\alpha^{(l)}$ and $\beta^{(l)}$ are learnable layer-specific fusion coefficients, enabling the model to adaptively adjust the emphasis on smoothness information versus discrepancy information according to task requirements and data characteristics.

### 3.3. Static Graph Representation Learning

The static branch is built upon the traditional graph convolutional operation, aiming to preserve the fundamental structural information of the original graph and prevent structural distortion caused by excessive dynamic adaptation. When the explicit connections in the graph are sufficiently accurate, the static branch provides reliable structural priors, effectively mitigating the performance bias that may arise from overfitting in the dynamic branch. Its feature propagation process is as follows:

$$\mathbf{H}_{stc}^{(l+1)} = \mathrm{ReLU}\left(\mathcal{A}^{low}\mathbf{H}_{stc}^{(l)}\mathbf{W}_{stc}^{(l)}\right) \qquad (8)$$

where $\mathbf{H}_{stc}^{(l)}$ denotes the feature representation at the $l$-th layer, $\mathbf{W}_{stc}^{(l)}$ is a learnable weight matrix, $\mathcal{A}^{low}$ is the low-pass filtering on the original graph adjacency $\mathbf{A}$. By modeling static structural information, this branch provides a stable and interpretable structural foundation for discriminative and comprehensive representation learning.

## 3.4. Multi-scale Representation Fusion

Shallow information propagation tends to capture local neighborhood features, while deeper propagation way explores more global topological structures. However, simply stacking various representations from different layers often fails to adequately extract discriminative features and may instead lead to information redundancy and overfitting. To fully integrate the multi-scale semantic information embedded across different layers, a multi-scale representation fusion mechanism within each channel is introduced. Taking the dynamic channel as an example, after $L$ layers of propagation, a set of intermediate representations is obtained, i.e., $\{\mathbf{H}_{dyn}^{(1)}, \mathbf{H}_{dyn}^{(2)}, \cdots, \mathbf{H}_{dyn}^{(L)}\}$. Rather than directly using the output of the final layer, a learnable weighted fusion over all layer-wise representations is performed, which serves as the final output of this channel:

$$\mathbf{H}_{dyn} = \gamma_{dyn}^{(0)}\mathbf{X} + \sum_{l=1}^{L} \gamma_{dyn}^{(l)}\mathbf{H}_{dyn}^{(l)} \qquad (9)$$

where $\{\gamma_{dyn}^{(l)}|l = 0, 1, \cdots, L\}$ is a set of learnable weight parameters, enabling the model to adaptively assign appropriate contributions to representations at different scales. The original feature representation $\mathbf{X}$ is also incorporated into the MsRF module, as it preserves fine-grained information and initial attributes that have not been smoothed by the propagation process, thereby enhancing the completeness and discriminability of the final representation. Similarly, for the static channel, with the propagation rule defined as Eq. (8), its final output is also obtained through multi-scale representation fusion:

$$\mathbf{H}_{stc} = \gamma_{stc}^{(0)}\mathbf{X} + \sum_{l=1}^{L} \gamma_{stc}^{(l)}\mathbf{H}_{stc}^{(l)} \qquad (10)$$

where $\{\gamma_{stc}^{(l)}|l = 0, 1, \cdots, L\}$ denotes another set of learnable multi-scale integration weight parameters.

## 3.5. Node Classification

The dynamic channel adaptively learns task-relevant latent semantic structures, while the static channel provides a stable and interpretable foundation of explicit structural information. To synergistically leverage the strengths of both, a learnable fusion parameter $\theta \in [0, 1]$ is generated via a sigmoid function, constructing the final node representation as an adaptively weighted combination of the outputs from the two channels:

$$\mathbf{H}_{fused} = \theta\mathbf{H}_{dyn} + (1 - \theta)\mathbf{H}_{stc} \qquad (11)$$

Finally, the final graph embedding representation $\mathbf{H}_{fused}$ is imported into a two-layer MLP classifier along with the

softmax function to obtain the probabilistic class distribution for each node:

$$\widehat{\mathbf{Y}} = \text{Softmax}(\text{MLP}(\mathbf{H}_{fused})) \qquad (12)$$

The element $\widehat{y}_{ic}$ denotes the predicted probability of the $i$-th node belonging to the $c$-th class. This forward propagation process of DCD-GNN, when combined with the loss function and backpropagation, forms a complete closed-loop training framework for model optimization. For the node classification task, the standard cross-entropy loss function over the labeled node set $\mathcal{V}_L$ is defined as follows:

$$\mathcal{L} = -\sum_{v_i \in \mathcal{V}_L} \sum_{c=1}^{C} y_{ic} \log(\widehat{y}_{ic}) \qquad (13)$$

where $C$ denotes the total number of classes, $y_{ic}$ represents the ground-truth label of node $v_i$, and $\widehat{y}_{ic}$ is the predicted probability that node $v_i$ belongs to class $c$.

## 3.6. Theoretical Analysis

Here, we provide simple theoretical analysis for DCD-GNN in alleviating the over-smoothing issue and complementary effect between static and dynamic representations.

**Alleviating over-smoothing issue in deep GNNs.** The over-smoothed embedding in deep GNNs mainly originates from the high-order neighboring aggregation in deeper layers, whose features are restricted to the low-frequency spectral subspace of the graph signal. By directly incorporating the original features $\mathbf{X}$ and the embedding representations from different layers into the final output, MsRF ensures that the final representation is no longer confined to the narrow subspace induced by deeply over-smoothed embeddings. Instead, it preserves complementary information from shallow local details and original attributes, effectively preventing representation collapse.

From the perspective of graph filtering, $\mathbf{H}_{dyn}^{(l)}$ can be regarded as a multi-hop aggregation on the initial features $\mathbf{X}$, i.e., $\mathbf{H}_{dyn}^{(l)} = \sum_{k=0}^{l} c_{l,k}\mathbf{S}^k\mathbf{X}$, where $\mathbf{S}$ is the propagation operator. Substituting this into Eq. (9) yields

$$\mathbf{H}_{dyn} = \sum_{l=0}^{L} \gamma_{dyn}^{(l)} \sum_{k=0}^{l} c_{l,k}\mathbf{S}^k\mathbf{X} = \sum_{k=0}^{L} \alpha_k\mathbf{S}^k\mathbf{X}, \qquad (14)$$

with $\alpha_k = \sum_{l=k}^{L} \gamma_{dyn}^{(l)}c_{l,k}$. This indicates that MsRF is equivalent to learning an adaptive polynomial filter that spans multiple neighborhood ranges. The learnable weights $\gamma_{dyn}^{(l)}$ allow the model to automatically adjust the receptive field scale according to the downstream task, rather than relying on a fixed propagation depth, thereby flexibly capturing structural information at various ranges.

**Complementarity of dual-channel fusion.** The complementarity of the dynamic and static channels is verified in the final fusion. Let $\hat{\mathbf{y}}_{dyn}$ and $\hat{\mathbf{y}}_{stc}$ denote the predictions of the dynamic and static channels, respectively, and let $\mathbf{y}$ be the ground-truth labels. The fused prediction is obtained by a learnable weight $\theta$ as

$$\hat{\mathbf{y}}_{fused} = \theta\hat{\mathbf{y}}_{dyn} + (1 - \theta)\hat{\mathbf{y}}_{stc} \tag{15}$$

Define the prediction errors of each channel as $\mathbf{e}_{dyn} = \hat{\mathbf{y}}_{dyn} - \mathbf{y}$ and $\mathbf{e}_{stc} = \hat{\mathbf{y}}_{stc} - \mathbf{y}$. The fusion error is then $\mathbf{e}_{fused} = \hat{\mathbf{y}}_{fused} - \mathbf{y} = \theta\mathbf{e}_{dyn} + (1 - \theta)\mathbf{e}_{stc}$. Its mean squared error can be expressed as

$$\mathbb{E}[\|\mathbf{e}_{fused}\|^2] = \theta^2\mathbb{E}[\|\mathbf{e}_{dyn}\|^2] + (1 - \theta)^2\mathbb{E}[\|\mathbf{e}_{stc}\|^2]$$
$$+ 2\theta(1 - \theta)\,\mathbb{E}[\langle\mathbf{e}_{dyn}, \mathbf{e}_{stc}\rangle] \tag{16}$$

When the errors of the dynamic and static channels are negatively correlated, i.e., $\mathbb{E}[\langle\mathbf{e}_{dyn}, \mathbf{e}_{stc}\rangle] \leq 0$, the overall fusion error becomes smaller than the weighted sum of the individual channel errors, thus bringing a performance gain.

In real-world graph data, such negative correlation naturally exists. The static channel strictly relies on the original topology; when the graph structure contains noise, missing edges, or exhibits heterophily, its predictions are prone to systematic bias. In contrast, the dynamic channel learns latent semantic correlations through the self-attention mechanism, which can correct misleading information, resulting in its error to being complementary to that of the static channel. Especially on heterophilic graphs, the dynamic channel not only supplements missing homophilic signals but also captures differentiating node characteristics through high-pass filtering, leading to an error pattern that is substantially different from, or even opposite to, that of the static channel. Hence, DCD-GNN adaptively increases the dynamic channel weight on heterophilic graphs, precisely exploiting the fusion gain brought by error negative correlation.

## 4. Experimental Results

Here, extensive experimental results are provided to verify the superiority of the proposed DCD-GNN, including quantitative classification performance compared to representative GNNs, ablation study, visualization results, etc.

### 4.1. Experimental Settings

**Datasets.** Eight widely used benchmark graph datasets are employed, including five heterophilic graphs: Chameleon, Wisconsin, Cornell, Texas (webpage networks), and Actor (co-occurrence network); as well as three homophilic citation networks: Cora, Citeseer, and Pubmed. In the citation networks, nodes represent papers and edges denote citation relationships, node attribute features are bag-of-words or

*Table 1.* The statistics of all used datasets.

| Dataset | #Nodes | #Features | #Classes | #Edges |
|---|---|---|---|---|
| Cora | 2708 | 1433 | 7 | 5429 |
| Citeseer | 3327 | 3703 | 6 | 4732 |
| Pubmed | 19717 | 500 | 3 | 44338 |
| Chameleon | 2277 | 2325 | 5 | 36101 |
| Texas | 183 | 1703 | 5 | 309 |
| Wisconsin | 251 | 1703 | 5 | 499 |
| Cornell | 183 | 1703 | 5 | 295 |
| Actor | 7600 | 931 | 5 | 33544 |

word frequency-inverse document frequency vectors, and labels correspond to the corresponding research domains. The webpage networks (Chameleon, Texas, Wisconsin, Cornell) model web pages as nodes and hyperlinks as edges, where the features are derived from page content. Actor is an actor co-occurrence network where edges indicate joint appearance on Wikipedia pages, and labels reflect actor categories based on Wikipedia content. The detailed statistical descriptions of all used datasets are provided in Table 1.

**Baseline Methods.** The performance of the proposed DCD-GNN is evaluated by comparing it with 19 state-of-the-art GNN methods in the node classification task. GCN (Kipf & Welling, 2017) performs spectral convolution via first-order neighborhood aggregation. GAT (Veličković et al., 2018) introduces attention mechanism to adaptively weight the neighboring nodes. GraphSAGE (Hamilton et al., 2017) enables inductive learning via neighbor sampling and aggregation. GCNII (Chen et al., 2020) employs residual connections with identity mapping to deepen networks. APPNP (Klicpera et al., 2019) uses personalized PageRank to preserve both local and global structural information. JKNet (Xu et al., 2018) alleviates over-smoothing through jump-knowledge connections that fuse multi-scale representations. Geom-GCN (Pei et al., 2020) incorporates geometric relationships to better capture structural patterns. OrderedGNN (Song et al., 2023) orders messages from different hops to handle heterophily. GPRGNN (Chien et al., 2021) learns adaptive weights via generalized PageRank. EvenNet (Lei et al., 2022) uses even-polynomial filters to enhance the robustness of representation learning. GNN-BC (Yang et al., 2022) learns different complementary representations for structure and attribute features via HSIC constraint and integrates them for final representation. Half-Hop (Azabou et al., 2023) inserts slow nodes into edges to refine message passing. PDE-GCN (Eliasof et al., 2021) models feature propagation as PDEs on manifolds to explore long-range correlations. NoSAF and NoSAF-D (Wang et al., 2024) employ node-specific neighboring aggregation layers for heterophilic graphs. LHS (Qiu et al., 2024) learns homophilic sub-structures for heterophilic graphs and im-

*Table 2.* Classification results on all datasets. The best classification results are in bold and the second best results are underlined. NA indicates that the reported method did not include results for this dataset, and the corresponding code is now unavailable.

| Dataset | Cora | Citeseer | Pubmed | Chameleon | Texas | Wisconsin | Cornell | Actor |
|---|---|---|---|---|---|---|---|---|
| MLP | 74.82 ± 2.22 | 70.94 ± 0.39 | 85.65 ± 0.25 | 46.59 ± 1.84 | 80.81 ± 4.75 | 85.29 ± 3.31 | 81.89 ± 6.40 | 40.18 ± 0.64 |
| GCN | 85.77 ± 0.20 | 73.68 ± 0.31 | 88.77 ± 0.24 | 28.18 ± 0.23 | 55.14 ± 5.16 | 51.76 ± 3.06 | 60.54 ± 5.30 | 30.43 ± 1.59 |
| GAT | 86.37 ± 0.30 | 74.32 ± 0.27 | 86.87 ± 0.62 | 42.93 ± 0.28 | 52.16 ± 6.63 | 49.41 ± 4.09 | 61.89 ± 5.05 | 29.26 ± 0.91 |
| GraphSage | 87.77 ± 1.04 | 71.09 ± 1.30 | 89.86 ± 0.38 | 49.24 ± 1.68 | 82.43 ± 6.14 | 81.18 ± 5.56 | 75.95 ± 5.01 | 40.24 ± 1.24 |
| GCNII | 88.49 ± 2.78 | 77.08 ± 1.21 | 88.47 ± 0.24 | 60.61 ± 2.00 | 77.57 ± 3.83 | 80.39 ± 3.40 | 77.86 ± 3.79 | 37.66 ± 1.10 |
| APPNP | 87.87 ± 0.85 | 76.53 ± 1.33 | 87.62 ± 0.44 | 54.30 ± 0.34 | 70.01 ± 1.59 | 70.72 ± 1.48 | 69.45 ± 0.99 | 37.05 ± 1.17 |
| JKNet | 88.93 ± 1.35 | 74.37 ± 1.53 | 87.65 ± 0.37 | 62.31 ± 2.76 | 83.78 ± 2.21 | 82.55 ± 4.57 | 75.68 ± 4.03 | 27.99 ± 1.29 |
| Geom-GCN | 85.19 ± 1.13 | 77.99 ± 1.23 | 90.05 ± 1.43 | 60.31 ± 1.77 | 57.58 ± 3.03 | 58.24 ± 4.33 | 56.76 ± 6.12 | 29.09 ± 0.93 |
| Ordered GNN | 88.37 ± 0.75 | 77.31 ± 1.73 | 90.15 ± 0.38 | 72.28 ± 2.29 | 86.22 ± 4.12 | 88.04 ± 3.63 | 87.03 ± 4.73 | 37.99 ± 1.00 |
| GPRGNN | 88.54 ± 0.67 | 80.13 ± 0.84 | 88.46 ± 0.31 | 67.49 ± 1.38 | 92.91 ± 1.32 | 91.71 ± 1.62 | 91.57 ± 1.96 | 34.63 ± 1.22 |
| EvenNet | 87.25 ± 1.24 | 78.65 ± 0.96 | 89.52 ± 0.31 | 67.57 ± 1.52 | 93.77 ± 1.73 | 93.55 ± 1.68 | 92.13 ± 1.71 | 40.48 ± 1.24 |
| GNN-BC | 88.75 ± 1.21 | 76.70 ± 0.77 | 88.13 ± 2.51 | 74.63 ± 0.93 | 85.01 ± 3.99 | 86.86 ± 3.88 | 85.14 ± 5.00 | 36.03 ± 1.00 |
| Half-Hop | 88.73 ± 1.22 | 80.33 ± 0.66 | 89.86 ± 0.36 | 62.98 ± 3.35 | 85.95 ± 6.42 | 87.59 ± 1.77 | 74.60 ± 6.06 | 36.82 ± 0.77 |
| PDE-GCN | 88.62 ± 1.03 | 79.98 ± 0.97 | 89.92 ± 0.38 | 66.01 ± 1.56 | 93.24 ± 2.03 | 92.85 ± 1.76 | 89.73 ± 1.35 | 39.76 ± 0.74 |
| NoSAF | 88.87 ± 1.14 | 78.05 ± 1.70 | 90.50 ± 0.47 | 70.31 ± 1.74 | 85.41 ± 6.14 | 88.20 ± 6.63 | 77.84 ± 6.72 | 38.56 ± 1.44 |
| NoSAF-D | 88.08 ± 1.52 | 77.31 ± 1.38 | 90.62 ± 0.47 | 69.14 ± 3.14 | 87.57 ± 7.77 | 89.40 ± 2.50 | 79.19 ± 7.21 | 38.76 ± 1.14 |
| LHS | 88.71 ± 0.70 | 78.53 ± 1.50 | NA | 72.31 ± 1.60 | 86.32 ± 4.50 | 88.32 ± 2.30 | 85.96 ± 5.10 | 38.87 ± 1.15 |
| DSF-Jacobi-I | 89.54 ± 0.19 | 78.18 ± 0.26 | 89.78 ± 0.09 | 74.88 ± 0.39 | 83.68 ± 1.12 | 85.34 ± 0.74 | 84.54 ± 0.81 | 39.64 ± 0.47 |
| DSF-Jacobi-R | 89.66 ± 0.19 | 78.23 ± 0.15 | 90.07 ± 0.10 | 75.00 ± 0.38 | 84.46 ± 0.81 | 86.13 ± 0.70 | 84.39 ± 0.88 | 40.02 ± 0.78 |
| DCD-GNN | **90.38 ± 1.38** | **82.43 ± 1.40** | **90.87 ± 0.54** | **78.36 ± 1.58** | **94.10 ± 2.34** | **98.13 ± 1.40** | **93.61 ± 2.48** | **41.04 ± 1.28** |

proves the accuracy of message passing. DSF-Jacobi-I and DSF-Jacobi-R (Guo et al., 2023) apply diverse spectral filters to capture both global and local patterns, resulting in comprehensive graph embedding representation.

**Implementation Details.** All experiments are conducted on a computer equipped with an NVIDIA RTX 4060 GPU. For performance evaluation, a unified splitting strategy is implemented across all datasets, where 60%, 20%, and 20% nodes are selected as training, validation, and testing sets, respectively. Node classification accuracy is adopted as the standard evaluation metric, serving as a community-accepted criterion that directly quantifies classification efficacy. Higher accuracy values reflect superior model performance in discriminative graph embedding representation learning. To ensure fair comparison, baseline results are derived from reported metrics in original publications. For partial comparison methods lacking published results on specific datasets, their publicly available implementations with comprehensive hyperparameter tuning are executed, and then the optimized performance metrics are subsequently reported under identical experimental conditions.

### 4.2. Node Classification

All comparison methods are evaluated on the above datasets, the mean accuracy and standard deviation over ten independent randomized runs are reported in Table 2. It can

be clearly observed that the proposed DCD-GNN method consistently outperforms all baseline methods across all datasets. DCD-GNN achieves an average improvement of 1.83% over the eight datasets, demonstrating its powerful superiority and robustness in complex real-world scenarios. The performance improvements are especially pronounced on heterophilic datasets, where an average improvement of 2.31% is observed. Notably, on the Wisconsin dataset, DCD-GNN outperforms the second-best PDE-GCN method by 5.28%, reaching near-perfect classification results. These experimental observations suggest that static information in heterophilic graphs often contains noise and interference, while modeling latent semantics and structures provides a more comprehensive correlation augmentation and correction. Traditional methods usually struggle in such complex graphs. Some representative GNN approaches (e.g., GCN, GAT, and GraphSAGE) even fall behind the MLP only utilizing attribute features on certain heterophilic datasets. Although some methods such as APPNP (which balances local and global information through personalized PageRank) and LHS (which improves structural stability) show some improvement over classic GNN methods, the performance improvements remain limited. The core reason is that these approaches rely too heavily on the pre-given static topological structure, failing to effectively model dynamic semantic correlations, and static topologies cannot adapt to evolving high-order semantic relationships. In contrast, DCD-GNN

maintains both the stability of static structures and the adaptability of dynamic structures, exhibiting strong robustness and generalization across diverse network scenarios. In addition, integrating self-attention learning into the spectral decomposition framework provides a fine-grained selection mechanism for diverse frequency information. Unlike traditional GNNs that act as fixed low-pass filters, the self-attention over multi-frequency signals allows for instance-specific frequency responses, effectively mitigating the over-smoothing problem by re-injecting high-frequency discriminative features and improving its representation capacity in complex and multi-scale relational graph. Consequently, DCD-GNN demonstrates significant advantages over representative GNN methods.

### 4.3. Ablation Study

To provide fine-grained analysis for the effectiveness of the proposed DCD-GNN, the contribution of each crucial component is evaluated through ablation experiments. Here, three ablation variants of DCD-GNN are designed: 1) **Variant-D** removes the dynamic semantic adjacency learning channel, which isolates the impact of latent semantics exploration. 2) **Variant-S** eliminates the static graph representation learning channel with the original topological structure. 3) **Variant-M** discards the multi-scale fusion module, relying entirely on the embeddings from the last layer for subsequent classification.

*Table 3.* The experimental results of three ablation modules and DCD-GNN, where the best classification results are bold.

| Dataset | Variant-D | Variant-S | Variant-M | DCD-GNN |
|---|---|---|---|---|
| Cora | 88.92 | 89.87 | 88.60 | **90.34** |
| Citeseer | 81.26 | 81.51 | 80.36 | **82.43** |
| Pubmed | 89.81 | 89.23 | 88.75 | **90.87** |
| Chameleon | 69.04 | 75.65 | 77.07 | **78.14** |
| Texas | 78.85 | 92.95 | 80.16 | **94.10** |
| Wisconsin | 77.00 | 97.25 | 92.75 | **98.13** |
| Cornell | 86.89 | 90.33 | 91.15 | **93.61** |
| Actor | 39.02 | 40.91 | 41.00 | **41.04** |

The experimental results of the ablation study are reported in Table 3. As shown in Table 3, all three variants are not superior to the complete DCD-GNN method, confirming the necessity and effectiveness of each component. In particular, Variant-D exhibits significant performance degradation in most cases, highlighting the crucial role of the dynamic channel in capturing adaptive latent semantics. The dynamically learned semantic structures provide additional and discriminative information for improving the power and robustness of representation learning. Furthermore, Variant-M also shows a clear performance decline, indicating that multi-scale information is essential for modeling complex high-order semantic relationships, and embeddings from

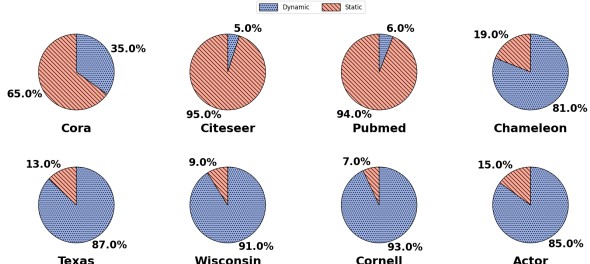

*Figure 2.* The distribution visualization of dynamic and static channel weights across different datasets.

different scales collectively enhance the final classification performance. Finally, the performance of Variant-S shows a relatively slight decrease. This indicates that the static channel effectively mitigates structural distortion caused by excessive dynamic adaptation, strikes a balance between latent semantics and static structure, and ensures that DCD-GNN learns more discriminative node representations.

*Table 4.* Classification accuracy under various network depths.

| Layer | 1 | 2 | 3 | 4 | 5 | 6 |
|---|---|---|---|---|---|---|
| Cora | 88.19 | 90.38 | 89.46 | 90.09 | 89.49 | 89.82 |
| Wisconsin | 98.13 | 96.88 | 93.25 | 94.00 | 95.38 | 92.50 |
| Texas | 93.28 | 94.10 | 92.46 | 93.77 | 93.27 | 93.61 |

### 4.4. Experimental Results of Over-smoothing

To verify the effectiveness of DCD-GNN in mitigating over-smoothing, we conduct experiments with varying network layers on three representative Cora, Wisconsin, and Texas datasets, where the results are shown in Table 4. Overall, the performance of DCD-GNN first improves significantly and then remains relatively stable as the number of layers increases. When the depth reaches 5 to 6 layers, DCD-GNN exhibits only mild performance degradation. This strongly demonstrates that the multi-scale representation fusion and dual-channel dynamic architecture endow DCD-GNN with a robust ability to maintain expressive power and alleviate the over-smoothing issue in deep GNNs.

### 4.5. Adaptive Weight Visualization

Here, the contribution differences of dynamic and static representation learning channels are shown by visualizing the corresponding weights, i.e., $\theta$ for dynamic channel and $1 - \theta$ for static channel. The weight visualization results on all datasets are presented in Figure 2. A significant phenomenon can be observed, that is, the dynamic channel accounts for a relatively lower weight in predominantly homophilic graphs, whereas its weight increases substantially in heterophilic graphs. This indicates that on homophilic graphs with relatively accurate connectivity, the static chan-

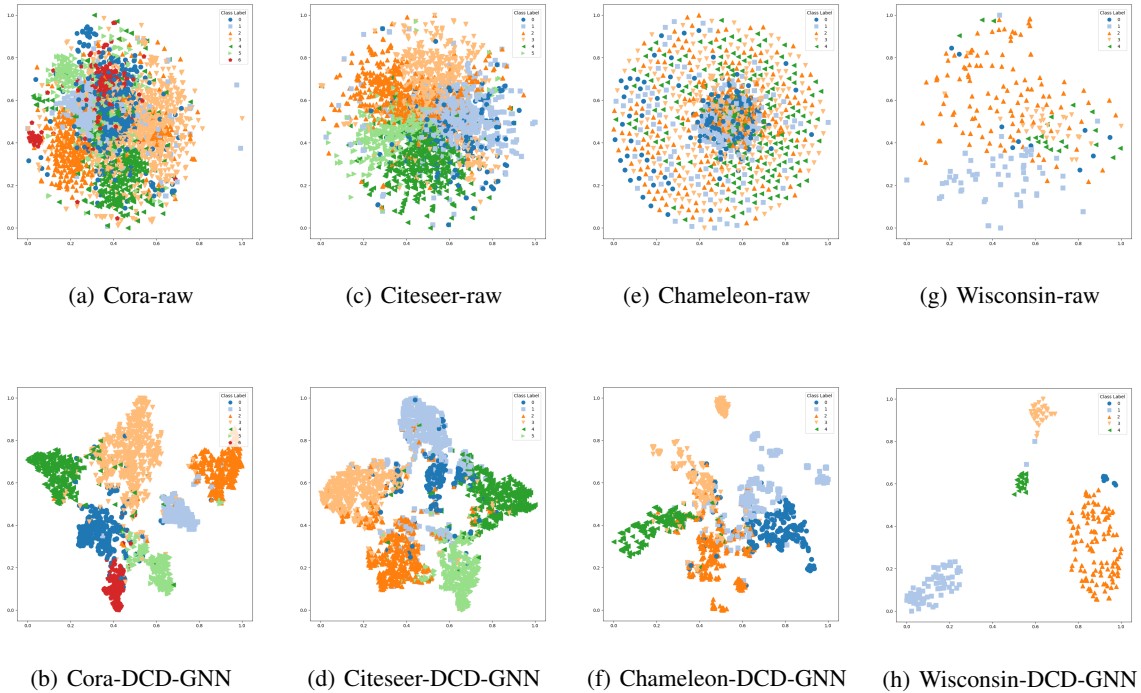

(a) Cora-raw      (c) Citeseer-raw      (e) Chameleon-raw      (g) Wisconsin-raw

(b) Cora-DCD-GNN    (d) Citeseer-DCD-GNN    (f) Chameleon-DCD-GNN    (h) Wisconsin-DCD-GNN

*Figure 3.* Visualization results of raw attribute feature and embedding representation learned by DCD-GNN on **Cora**, **Citeseer**, **Chameleon**, and **Wisconsin** datasets.

nel provides reliable topological priors, and the dynamic channel only needs to supply moderate refinement. In contrast, on graphs with higher noise and stronger heterophilic connections, the DCD-GNN tends to assign greater weight to the dynamic channel, enabling it to better capture latent semantic structures and thereby correct the representation bias introduced by noise in the original adjacency matrix. And, the high-pass filtering in the dynamic channel also provides beneficial representation for heterophilic graphs. This observation confirms that DCD-GNN can adaptively balance static structural priors with dynamic semantic inference according to the intrinsic properties of the graph data, effectively enhancing both the robustness and discriminability of node representations.

### 4.6. Embedding Representation Visualization

To validate the effectiveness of the proposed DCD-GNN model in discriminative representation learning, the final embedding representation $\mathbf{H}_{fused}$ is visualized by t-SNE (Van der Maaten & Hinton, 2008) in the 2-D space. Experiments on four representative datasets, comprising two homophilic and two heterophilic graphs, are shown to comprehensively examine the model's performance across different graph structures. The visualization results are presented in Figure 3, where distinct shapes and colors correspond to different node classes. These visualization results

reveal several key observations: node distributions based on raw attribute features exhibit considerable overlap, with blurred inter-class boundaries and noticeable category mixing. In contrast, the embeddings learned by DCD-GNN display clear inter-class separability and high intra-class compactness. These findings provide compelling visual evidence that DCD-GNN can significantly enhance the discriminability of the embeddings, thereby establishing a stronger representational foundation for downstream tasks.

## 5. Conclusion

This paper proposed the Dual-Channel Dynamic Graph Neural Network (DCD-GNN), a framework that addresses the limitations of existing GNNs in modeling latent semantic structures by designing both a static structure-preserving channel and a dynamic adjacency-enhancing channel. The DCD-GNN adaptively fused static topological priors with dynamic semantic inference, learning node representations with stronger discriminability and improving the representation robustness. Ablation studies confirmed the necessity of each crucial module, and extensive experimental results on real-world graph datasets demonstrated that the proposed DCD-GNN consistently achieves stable and superior classification performance across various types of graph data.

## Acknowledgment

This research was supported by the National Natural Science Foundation of China under Grant 62225303, 62403043, and 62433004; in part by the Beijing Natural Science Foundation under Grant L2603012; in part by the Interdisciplinary Research Center of Beijing University of Chemical Technology under Grant XK2025-06.

## Impact Statement

This paper presents work whose goal is to advance the field of Machine Learning. There are many potential societal consequences of our work, none which we feel must be specifically highlighted here.

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
