# OpenReview forum: "Dual-channel Dynamic Graph Neural Networks with Adaptive Adjacency Learning and Multi-scale Representation Fusion"
_ICML.cc/2026/Conference — ICML 2026 regular_

### Official Review · Reviewer_ySjr · 2026-03-06

**Soundness:** 3
**Presentation:** 4
**Significance:** 3
**Originality:** 4
**Overall Recommendation:** 5
**Confidence:** 5

**Summary:**

This paper introduces DCD-GNN, a model that enhances graph neural networks by adding a "dynamic channel" to the standard "static channel" based on the input graph. The dynamic channel uses self-attention to adaptively create connection edges, and it processes them through separate low-pass and high-pass filters. The representations from multiple layers of both channels are combined using learned weights. The final node representation is a weighted sum of the two channel outputs. The DCD-GNN is tested on standard citation and webpage datasets, showing improvements over a long list of baselines.

**Compliance With Llm Reviewing Policy:**

Affirmed.

**Final Justification:**

The author provided detailed replies to my concerned issues, which improved my understanding of the contribution of the paper. This paper is innovative and complete. The author should further improve the grammatical description in the future. I decided to raise my score, i.e., 5.

**Key Questions For Authors:**

1. Dynamic semantic learning is interesting at both high and low frequencies. Does the dynamic adjacency introduce semantic shortcuts that could harm structural consistency.

2. What is the difference between dynamic learning in high frequency and low frequency, from the physical meaning and form ( sparsity or dense patterns)

3. Authors should analyze the computational complexity and its scalability on large-scale datasets, particularly in terms of training and inference efficiency.

4. The universality and extensibility of the model are important. Can the proposed DCD-GNN framework naturally extend to temporal or dynamic graphs in the open environment.

5. Is the model robust to graph sparsity variations.

**Limitations:**

I suggest that authors should discuss its universality and extensibility of the proposed method, especially for temporal or dynamic graphs in the open environment.

**Strengths And Weaknesses:**

**Soundness**: The experimental design is comprehensive and the results are convincing, particularly as the performance improvements across multiple benchmark datasets are both statistically significant and consistently stable.

**Presentation**: The paper is well-structured, but readability could be improved by more clearly distinguishing the "what" and "why" from the "how." The tables and figures are excellent and easy to understand.

**Significance**: The contributions of this paper are solid. It demonstrates that a carefully designed dual-channel architecture combined with frequency decomposition can significantly improve performance on complex graphs.

**Originality**: The originality is satisfied. The issues addressed in this paper are important, and the proposed method is novel. However, some design motivations require theoretical verification.

---

> ### Author Rebuttal · Authors · 2026-03-30
>
> **Q1**: The dynamic adjacency matrix learns implicit semantic associations from node features through the self-attention mechanism, which does introduce new edges that may be inconsistent with the original graph structure. However, our design avoids disrupting structural consistency through two mechanisms. First, the dynamic channel does not completely replace the original structure; instead, it combines the original adjacency matrix with the learned dynamic adjacency matrix via convex combination, preserving the original topological prior. Second, the static channel independently maintains information propagation based on the original graph structure, ensuring that fundamental topological information is not lost. The outputs of the two channels are adaptively fused through a learnable parameter $\theta$, allowing the model to balance dynamic semantics and static structure according to task requirements. Experiments demonstrate that this design enhances performance without causing structural distortion. Furthermore, we observe that the dynamic channel is assigned higher weight in heterophilic graphs, indicating that the model tends to introduce semantic information only when necessary, rather than arbitrarily, thereby avoiding the negative impact of semantic shortcuts.
>
> **Q2**: In the dynamic channel, we construct a dynamic low-pass filtering adjacency matrix and a dynamic high-pass filtering adjacency matrix, respectively. The dynamic low-pass filtering adjacency matrix emphasizes similarity between nodes and aims to aggregate locally smooth information, which is more suitable for homophilic relationships. It typically results in a relatively dense connectivity pattern, as similar nodes tend to be connected to one another. The dynamic high-pass filtering adjacency matrix, in contrast, emphasizes differences and captures local variations, which is more suitable for heterophilic relationships, and its connectivity pattern may be sparser. From a physical perspective, low-pass filtering corresponds to the low-frequency components of graph signals, reflecting global consistency, while high-pass filtering corresponds to high-frequency components, reflecting local variations and node-specific information.
>
> **Q3**: We conducted a computational complexity analysis of DCD-GNN, revealing that its core computational cost primarily stems from the self-attention mechanism in the dynamic channel, with a overall complexity of $\mathcal{O}(N^2d)$. Experiments confirm that training on medium to large-scale datasets, such as Pubmed, already incurs significant overhead, making direct scalability to large-scale graphs challenging. Therefore, reducing computational cost while maintaining model performance will be a key focus of our future research. One potential direction is to replace the current node-to-node global self-attention mechanism with a node-to-cluster local self-attention mechanism.
>
> **Q4**: The core idea of DCD-GNN is dual-channel dynamic graph learning, in which the dynamic channel can adaptively adjust adjacency relationships based on node features. We believe this mechanism can be naturally extended to temporal graph or dynamic graph scenarios. For dynamic graphs, where the graph structure may change at each time step, we can treat the graph at each time step as a snapshot and apply DCD-GNN to each snapshot, while incorporating cross-time information propagation, such as RNNs or attention mechanisms, to capture temporal dependencies. Additionally, the self-attention mechanism in the dynamic channel can share parameters across different time steps or model temporal relationships between nodes through temporal self-attention. The static channel can preserve the original structure at each time step, ensuring stability. Therefore, the framework of DCD-GNN offers good scalability, and its application to dynamic graphs can be explored in future research.
>
> **Q5**: We experimentally validated the robustness of the model to graphs with varying sparsity. In our experiments, we used datasets ranging from sparse graphs, such as Cora with 5,429 edges, to relatively dense graphs, such as Pubmed with approximately 44,338 edges. DCD-GNN achieved the best performance across all datasets, demonstrating its insensitivity to variations in graph sparsity. Notably, on sparse heterophilic graphs, such as Texas with only 309 edges, the dynamic channel is able to supplement missing semantic connections through self-attention, thereby improving performance. The dynamic channel can learn strongly correlated latent edges from node features, mitigating the issue of insufficient information caused by structural sparsity.

---

> > ### Author Rebuttal · Reviewer_ySjr · 2026-04-02
> >
> > The author provided detailed replies to my concerned issues, which improved my understanding of the contribution of the paper. This paper is innovative and complete. The author should further improve the grammatical description in the future. I decided to raise my score, i.e., 5.

---

> > > ### Author Response · Authors · 2026-04-04
> > >
> > > It is a great honor that our reply has solved your concerns about our paper. Thank you for your recognition of our study and improving your score. This is very meaningful for us. We sincerely thank you once again for your meticulous review and the valuable suggestions to improve our work.

---

### Official Review · Reviewer_2Uhd · 2026-03-07

**Soundness:** 3
**Presentation:** 3
**Significance:** 4
**Originality:** 4
**Overall Recommendation:** 5
**Confidence:** 4

**Summary:**

A Dual-Channel Dynamic Graph Neural Network (DCD-GNN) is proposed, which combines a static structure-preserving channel with a dynamic adjacency-learning channel. The authors intend to outline a central concept of separating stability and adaptability in graph representation learning, where static topology ensures structural consistency while dynamic self-attention-based adjacency learning captures latent semantic relations. The dynamic channel further decomposes information propagation into low-pass and high-pass filtering branches, followed by adaptive fusion. Multi-scale representation fusion is applied in both channels, and their outputs are adaptively integrated for node classification. Overall, the main challenge explored by the proposed DCD-GNN is how to balance explicit graph structure with latent semantic relationships in a unified and stable learning framework.

**Compliance With Llm Reviewing Policy:**

Affirmed.

**Final Justification:**

The authors’ response has addressed my concerns, and I will maintain my score.

**Key Questions For Authors:**

(1)How does the proposed DCD-GNN method alleviate the over-smoothing issue, for both static and dynamic channels, as well as for the overall model.

(2)In the DGRL module, the self-attention learning is utilized for both low-passing and high-passing, what are the advantages of full attention compared to k-nearest neighbors?

(3)The author needs to experimentally verify the advantages of the proposed DCD-GNN in mitigating over-smoothing.

(4)The sigmoid function generates a single scalar $\theta$ for fusing the two channels. Would a node-level or feature-level fusion parameter provide more fine-grained control?

**Limitations:**

No. The computational scalability may be worth paying attention to in the future.

**Strengths And Weaknesses:**

Str1: The decoupling and then merging of static and dynamic channel is a logical and interesting idea. The specific design of the dynamic channel with high/low-frequency decomposition via self-attention is a novel contribution.

Str2: The concerned problem is important, and the results are strong, particularly on heterophilic graph representation learning. Its significance is high.

Wea1: Although this paper is clearly written, the methodology section is quite dense and could be improved by incorporating more transitional text to guide readers through the purpose of each equation.

Wea2: The authors could enhance reader understanding of the optimization process by discussing the convergence properties of the dual-channel learning procedure in more detail.

---

> ### Author Rebuttal · Authors · 2026-03-30
>
> **Q1**: The over-smoothing problem is a common issue in deep GNNs, where node representations tend to become indistinguishable. DCD-GNN alleviates over-smoothing through the following well-designed mechanisms: 1) The high-pass filtering adjacency matrix in the dynamic channel emphasizes node differences and preserves high-frequency information, helping to avoid excessive smoothing and protecting discriminative node-specific information. 2) The multi-scale fusion module aggregates representations from all layers rather than using only the final layer, allowing shallow local information to be retained and thus mitigating the effects of deep-layer smoothing. 3) The dual-channel fusion balances dynamic and static information through a learnable parameter $\theta$, where dynamic information provides diversity search space and static information provides training and learning stability. The above three mechanisms synergistically alleviate the over-smoothing issue of proposed DCD-GNN. As the following experiments show (**Rebuttal for Q3**), when the number of layers increases, the performance of the DCD-GNN does not show a cliff like decline, and improves in partial datasets.
>
> **Q2**: The main advantage of using full attention over kNN is that full attention can capture global, non-local semantic relationships, whereas kNN only considers local neighborhoods. In many graph datasets, important implicit relationships may exist between distant nodes, and full attention can model such long-range dependencies. Additionally, full attention produces soft weights through softmax function, which are differentiable and amenable to end-to-end training, while kNN typically requires discrete selections that are difficult to integrate into gradient descent. Although full attention incurs higher computational cost, it offers stronger expressiveness and is particularly well-suited for capturing complex semantics. For large-scale graphs, we can combine sparse attention to approximate full attention, balancing efficiency and effectiveness. This is mainly due to the three crucial designs in DCD-GNN, as analyzed in **Rebuttal for Q1**.
>
> **Q3**: According to your suggestions, we conducted over-smoothing experiments on three representative benchmark datasets. As shown in the table below, on the whole, as the number of layers increases, the performance of DCD-GNN first increases significantly. Although DCD-GNN also exhibits some degree of performance degradation as the number of layers increases (5-6 layers), the decline is noticeably more gradual, showing only mild attenuation. However, traditional GNNs typically achieve their best performance at 2 to 3 layers, after which performance drops sharply. This experimental result strongly demonstrates the effectiveness of DCD-GNN in mitigating the over-smoothing problem.
>
> | Layer | 1 | 2 | 3 | 4 | 5 | 6 |
> | :--- | :---: | :---: | :---: | :---: | :---: | :---: |
> | Cora | 88.19 | 90.38 | 89.46 | 90.09 | 89.49 | 89.82 |
> | Wisconsin | 98.13 | 96.88 | 93.25 |94.00 | 95.38 | 92.50 |
> | Texas | 93.28 | 94.10 | 92.46 | 93.77 | 93.27 | 93.61 |
>
> **Q4**: Using a single scalar $\theta$ is indeed a simplified model design, but we have validated its performance effectiveness. However, node-level or feature-level fusion may provide fine-grained importance adjustment. For example, each node could adjust the ratio of dynamic and static information according to its own characteristics, or each feature dimension could be fused independently. Given that the self-attention mechanism already incurs a complexity of $\mathcal{O}(N^2d)$, such fine-grained attention approaches would undoubtedly further increase computational complexity and training difficulty. Feature-level fusion, in particular, may be overly complex. We believe the current design strikes a balance between computational efficiency and practical performance, but finer-grained fusion strategies could be explored in the future, especially in graphs with strong node heterogeneity.
>
> **W1**: In the future, we will provide a more logical transitional description of method design to improve readability and facilitate  strong understanding for readers.
>
> **W2**：The dual-channel framework is trained end-to-end with standard gradient-based optimization. Since the final prediction is a differentiable fusion of the dynamic and static channels, the whole model is optimized jointly under a unified loss, rather than through alternating or adversarial training. The fusion function $\theta$ is smooth and differentiable, enabling stable gradient flow to both channels. Therefore, its optimization follows the standard convergence behavior of non-convex deep models, i.e., convergence to a stationary point under routine conditions. **The experimental convergence results are shown in https://anonymous.4open.science/r/DCD-GNN-2BB1/loss_chameleon.png**. It can be observed that the training and optimization process of this method is convergent and stable.

---

> > ### Author Rebuttal · Reviewer_2Uhd · 2026-04-01
> >
> > The authors’ response has addressed my concerns, and I will maintain my score.

---

> > > ### Author Response · Authors · 2026-04-04
> > >
> > > Thank you for your recognition of our study. We sincerely thank you once again for your meticulous review and the valuable suggestions to improve our work.

---

### Official Review · Reviewer_3dka · 2026-03-10

**Soundness:** 3
**Presentation:** 3
**Significance:** 3
**Originality:** 3
**Overall Recommendation:** 4
**Confidence:** 4

**Summary:**

This study proposes DCD-GNN, a new graph representation learning framework that learns from both the given graph structure and a dynamically inferred semantic structure. The dynamic channel uses self-attention to create new edges based on feature similarity, separated into low-frequency and high-frequency components. The outputs from multiple layers of both channels are then combined using learned weights before being fused for the final node classification task. Its core idea is to learn dynamic adaptability and structural stability separately through a dual-channel parallel architecture, enabling optimal fusion of multi-level and multi-scale structural information. The proposed method is evaluated on a mix of homophilic and heterophilic datasets, showing consistent improvements over many existing baselines.

**Compliance With Llm Reviewing Policy:**

Affirmed.

**Final Justification:**

My concerns have been adequately addressed. I maintain my initial rating.

**Key Questions For Authors:**

1. DCD-GNN only explores high-passing and low-passing information in the dynamic channel, not in the static channel. Please explain this.

2. In the DGRL module, the dynamic low-passing and high-passing information are first fused and then utilized as the input of dynamic correlation exploration and representation learning in the next layer. What are the advantages of this design? Why not stack multiple layers in low-frequency and high-frequency channels and then fuse them in the final layer?

3. The performance of DCD-GNN on Wisconsin is exceptionally high (98.13%) and the improvement is clear. Is there a specific property of this dataset that makes it particularly well-suited for DCD-GNN?

4. Using only dynamic channels achieve such strong performance on Texas and Wisconsin in the ablation study. What is the reason behind it and what is the connection with the innovation of this study?

5. Would task-specific tuning of $\theta$ significantly affect the node classification performance?

6. Does dynamic adjacency learning introduce the training instability?

**Limitations:**

This study does not explicitly discuss the interpretability of learned dynamic structures.

**Strengths And Weaknesses:**

1. The specific framework within a dual-channel and dual-frequency strategies is novel and demonstrably effective for general graph representation learning.

2. The paper is generally clear and well-organized. The methodology section is detailed enough to be reproducible and easy to understand. The writing is concise and to the point. The tables and figures effectively present the key results and provide persuasive experimental verification for proposed method.

3. Although the proposed method is interesting and innovative, the theoretical analysis needs to be improved.

---

> ### Author Rebuttal · Authors · 2026-03-30
>
> **Simple Theoretical Analysis**:
> Over-smoothing causes deep embeddings to gradually collapse into a low-dimensional subspace, reducing node discriminability. By fusing raw features and multi-scale representations $\mathbf{H}_{dyn}= \sum\_{l=0}^L \gamma\_{dyn}^{(l)} \mathbf H^{(l)}$, final representation is not restricted to subspace induced by deep over-smoothed embeddings alone. Instead, it preserves complementary information from shallow layers and input features, thus mitigating representation collapse.
>
> Learnable weights $\gamma^{(l)}$ allow model to automatically select optimal scale according to task. From an information-theoretic perspective, different layers capture neighborhood information at different ranges: $\mathbf{H}^{(l)}$ contains aggregated features within $l$-hop neighborhood. MsRF is equivalent to learning an adaptive polynomial filter:
>
> $$\mathbf{H}_{dyn}= \sum\_{l=0}^L \gamma\_{dyn}^{(l)} \mathbf H^{(l)} =  \sum\_{l=0}^L \gamma\_{dyn}^{(l)} \left(\sum\_{k=0}^l c\_{l,k}(\mathbf S) ^k \mathbf X\right) = \sum\_{k=0}^L \alpha\_k\mathbf S^k \mathbf X$$
>
> where $c_{l, k}$ are determined by mixing coefficients in iterative process, $\alpha\_k = \sum\_{l=k}^L\gamma\_{dyn}^{(l)}c\_{l,k}$. Hence, MsRF can be interpreted as learning an adaptive polynomial filter over multiple neighborhood ranges.  Since $\mathbf H^{(l)}$ aggregates information from at most the $l$-hop neighborhood, learnable coefficients $\gamma\_{dyn}^{(l)}$, allow model to adaptively emphasize the most task-relevant receptive-field scales, rather than relying on a fixed propagation order.
>
> Final fusion weight $\theta$ is also learned automatically by network and acts as an adaptive ensemble coefficient between dynamic and static channels. Let $e\_{dyn} = \hat{y}\_{dyn} - y$, $e\_{stc} = \hat{y}\_{stc} - y$ denote prediction errors of dynamic and static channels, respectively, for fused prediction, $\hat{y}_{fused} = \theta \hat{y}\_{dyn} + (1-\theta)\hat{y}\_{stc}$, corresponding error is $e\_{fused} = \hat{y}\_{fused} - y = \theta e\_{dyn}+ (1-\theta)e\_{stc}$. Its mean squared error is
>
> $$\mathbb{E}[e\_{fused}^2] = \theta^2 \mathbb E[e\_{dyn}^2]+(1-\theta)^2\mathbb E[e\_{stc}^2]+2\theta(1-\theta)\mathbb E[e\_{dyn}e\_{stc}]$$
>
> When two channels produce negatively correlated errors, i.e., $\mathbb E[e\_{dyn}e\_{stc}]<0$ , cross term becomes negative and reduces overall fusion error. This explains why combining two channels can outperform either one alone. Empirically, we observe that $\theta$ tends to be larger on heterophilic graphs and smaller on homophilic graphs, indicating that DCD-GNN successfully exploits complementarity between dynamic and static channels in different structural regimes.
>
> **Q1**: static channel maintains stability via standard graph convolution (a low-pass filter). Our design has a clear division: the dynamic channel explores full-spectrum information, while the static channel preserves the original topology and avoids overcomplication. Experimental results also show the dynamic channel alone improves performance, and the static channel provides a stable foundation.
>
> **Q2**: Each layer fuses low-pass and high-pass information before passing it to the next layer. This enables cross-frequency interaction at every step, enriching representations and progressively building complex semantics. Separately stacking multiple layers would risk independent evolution, low-pass becoming oversmooth, high-pass overly noisy. Early fusion allows mutual regulation and collaboration, improving the representation discriminability.
>
> **Q3**: Wisconsin is a heterophilic graph with complex feature-label relationships. the dynamic channel of DCD-GNN learns implicit homophilic and heterophilic structures from features, compensating for limited original edges. Its high-dimensional features enable self-attention to capture latent semantics. The static channel retains useful structural information, and multi-scale fusion captures multi-range dependencies. These factors yield strong performance.
>
> **Q4**: In ablation study, DCD-GNN-S (dynamic only) performs well on Texas and Wisconsin, close to the full model. This indicates the dynamic channel drives performance on heterophilic graphs. It validates our core innovation: compensating static limitations via dynamically learned implicit structures. However, the full model still outperforms, showing the static channel still helps.
>
> **Q5**: $\theta$ is a learnable parameter generated by sigmoid function, which is automatically adjusted during training without manual tuning. With a fixed initialization, the model adaptively learns a suitable $\theta$ for various task, eliminating task-specific tuning.
>
> **Q6**: Dynamic adjacency learning can cause instability because self-attention learns global dependencies with varying gradients. We stabilize by: 1) dividing by $\sqrt{d}$ in self-attention to control gradient scale; 2) incorporating the static channel for stable gradient flow.

---

> > ### Author Rebuttal · Reviewer_3dka · 2026-04-03
> >
> > I thank the authors for the rebuttal. All my concerns are well addressed.

---

> > > ### Author Response · Authors · 2026-04-04
> > >
> > > We sincerely thank you once again for your meticulous review and the valuable suggestions to improve our work.

---

### Official Review · Reviewer_K829 · 2026-03-11

**Soundness:** 3
**Presentation:** 4
**Significance:** 3
**Originality:** 3
**Overall Recommendation:** 4
**Confidence:** 5

**Summary:**

This paper introduces the Dual-Channel Dynamic Graph Neural Network (DCD-GNN), a novel framework designed to overcome the limitations of traditional GNNs that rely on static and often imperfect graph structures. The core contribution is a parallel architecture comprising a dynamic channel, which uses self-attention to learn latent semantic correlations across high-passing and low-passing frequencies, and a static channel, which preserves explicit topological stability. A multi-scale representation fusion mechanism is also proposed to adaptively combine graph embeddings from different layers. Experiments on eight benchmark datasets demonstrate that DCD-GNN achieves state-of-the-art performance, particularly on heterophilic graphs.

**Compliance With Llm Reviewing Policy:**

Affirmed.

**Final Justification:**

Thank you for the thoughtful and detailed rebuttal. The authors have adequately addressed my concerns, and I appreciate the clarifications. I will keep my current score.

**Key Questions For Authors:**

Q1: Do the two adjacency matrices used in dynamic channel learning consistently capture different types of relationships?

Q2: Do the multi-scale fusion parameters exhibit consistent patterns across different datasets during training? For instance, do shallow layers tend to have higher weights in homophilic graphs compared to homophilic ones?

Q3: What is the effect of multi-scale information fusion on improving the universality of DCD-GNN on homophilic and homophilic graphs? Especially from dynamic and static channels.

Q4: Could you provide more details on the computational complexity of the two dynamic adjacency matrices compared to a standard GCN layer?

Q5: How sensitive is the model to the choice of $\lambda$ when blending static and dynamic adjacency across different datasets?

**Limitations:**

This paper mainly focuses on its contributions and does not overstate its limitations. This paper needs to discuss its scalability on large-scale graph.

**Strengths And Weaknesses:**

Strengths:

S1: The proposed methodology is technically robust and solid. And, this paper is exceptionally well-structured and clearly written.

S2: This work addresses a critical and relevant problem in graph representation learning. By effectively integrating latent semantics with explicit structure, it offers a significant advancement for complex real-world applications. The improvements on heterophilic datasets are particularly noteworthy and could influence future research directions in this area.

S3: This paper clearly distinguishes itself from prior work with its own unique contributions, and its innovation is relatively persuasive.

Weaknesses:

W1: Some experimental results need to be further enriched. For example, visualization experiments need to increase the results of compared methods.

W2: Some method details need to be provided to improve readability.

---

> ### Author Rebuttal · Authors · 2026-03-30
>
> **Q1**: Yes, the dynamic low-pass and high-pass adjacency matrices we designed aim to capture different types of graph relationships. The dynamic low-pass matrix emphasizes latent homophilic semantics in the low-pass sub-channel by aggregating similar nodes, while the dynamic high-pass matrix highlights differences to capture latent heterophilic semantics in the high-pass sub-channel. In the fan chart visualization of the parameter analysis, we observed that the high-pass sub-channel contributes more significantly in heterophilic graphs, whereas the low-pass sub-channel plays a more important role in homophilic graphs. This indicates that the two variour matrices indeed capture different types of relationships, playing different but complementary roles and thus improving the generality of the model in both homophilic and heterophilic graphs.
>
> **Q2**: We conducted additional multi-scale parameter analysis experiments to examine the parameter learning results of the proposed DCD-GNN on different datasets. We selected two representative homophilic datasets (Cora and Citeseer) and two representative heterophilic datasets (Chameleon and Wisconsin). After normalization, we found that the information proportions of dynamic channels at layers 1 and 2 are 0.22 and 0.20, respectively, on Cora; 0.22 and 0.19 on Citeseer; 0.16 and 0.32 on Chameleon; and 0.27 and 0.27 on Wisconsin. These results indicate that the DCD-GNN provides greater emphasis on dynamic adjacency learning in heterophilic graphs, which is consistent with theoretical analysis, suggesting that a consistent learning pattern exists across different datasets. For datasets with varying degrees of heterophily, multi-scale fusion enables effective adaptive selection. More detailed results are shown in the table below.
>
> | Layer | 1 | 2 | 3 | 4 | 5 |
> | :--- | :---: | :---: | :---: | :---: | :---: |
> | Cora | 0.22 | 0.20 | 0.20 | 0.20 | 0.20 |
> | Citeseer | 0.22 | 0.19 | 0.19 | 0.19 | 0.19 |
> | Chameleon | 0.16 | 0.32 | 0.20 | 0.15 | 0.16 |
> | Wisconsin | 0.27 | 0.27 | 0.18 | 0.14 | 0.14 |
>
> **Q3**: The multi-scale fusion module enables the model to simultaneously leverage representations from different propagation depths, thereby better adapting to various graph types. In homophilic graphs, shallow representations already contain sufficient information, while deep layers may introduce over-smoothing; in heterophilic graphs, deep layers can capture long-range dependencies, compensating for the insufficiency of local information. Through multi-scale fusion in both the static and dynamic channels, the model can adjust scale weights according to the characteristics of each channel. Experimental results show that removing the multi-scale fusion module leads to a significant performance drop, particularly pronounced in heterophilic graphs (e.g., Texas decreases from 94.10% to 80.16%), indicating that multi-scale fusion is crucial for heterophilic graphs. Meanwhile, multi-scale fusion in the static channel helps maintain structural stability, while multi-scale fusion in the dynamic channel enhances semantic richness; their combination improves the model's generalizability.
>
> **Q4**: The computational complexity of a standard GCN layer is $\mathcal{O}(N^2d)$. In the dynamic channel, we first learn the adjacency matrix through self-attention computation, with a complexity of $\mathcal{O}(N^2d)$. Then, we construct the low-pass and high-pass dynamic matrices separately, where the complexity of the low-pass dynamic matrix is $\mathcal{O}(N^{2})$, and that of the high-pass dynamic matrix is also $\mathcal{O}(N^{2})$. Therefore, the complexity of the dynamic channel is dominated by $N^{2}$, while the complexity of the static channel is also $\mathcal{O}(N^2d)$. Hence, the theoretical computational complexity of our method is consistent with the classical GCN.
>
> **Q5**: We use a hyperparameter $\lambda$ to control the mixing ratio between the original adjacency matrix and the dynamically learned adjacency matrix, and we conduct sensitivity experiments on $\lambda$. Here, we also select two representative homophilic datasets and two representative heterophilic datasets. As shown in the table below, within a reasonable range, homophilic datasets are not particularly sensitive to variations in $\lambda$, whereas heterophilic datasets exhibit greater sensitivity. This indicates that the underlying patterns in heterophilic datasets are more complex, and appropriate supplementation of dynamic information is crucial, it is necessary to avoid both insufficient information that fails to fully capture potential patterns and excessive information that leads to redundancy.
>
> | $\lambda$ | 0.1 | 0.3 | 0.5 | 0.7 | 0.9 |
> | :--- | :---: | :---: | :---: | :---: | :---: |
> | Cora | 89.15 | 89.67 | 89.57 | 90.38 | 89.98 |
> | Citeseer | 81.31 | 82.40 | 82.10 |82.43 | 81.58 |
> | Chameleon | 68.71 | 74.40 | 77.02 | 78.36 | 76.37 |
> | Wisconsin | 90.12 | 89.88 | 93.63 | 98.13 | 97.38 |

---

> > ### Author Rebuttal · Reviewer_K829 · 2026-04-03
> >
> > Thank you for providing serious rebuttal to my concerned questions and providing a detailed response. I maintain the original score, but the paper is generally acceptable.

---

> > > ### Author Response · Authors · 2026-04-04
> > >
> > > It is a great honor that our reply has solved your concerns about our paper. Thank you for your recognition of our study. We sincerely thank you once again for your meticulous review and the valuable suggestions to improve our work.

---

### Decision · Program_Chairs · 2026-04-30

**Decision:**

Accept (regular)

**Comment:**

This paper proposes a dual-channel architecture to balance structural stability and adaptive semantic modeling. The reviewers unanimously recognized the novelty of the proposed method, which combines dynamic adjacency learning with frequency decomposition and multi-scale fusion. The proposed method also shows superior performance in experiments, especially on heterogeneous graphs. While some reviewers raised some concerns about the rationale, scalability, and plausibility of some design choices, the authors addressed their concerns with detailed responses in their responses, and the reviewers maintained positive scores.